# Self-Selected Pacing During a World Record Attempt in 40 Ironman-Distance Triathlons in 40 Days

**DOI:** 10.3390/ijerph17072390

**Published:** 2020-04-01

**Authors:** Caio Victor Sousa, Rinaldo Wellerson Pereira, Thomas Rosemann, Pantelis T. Nikolaidis, Beat Knechtle

**Affiliations:** 1College of Arts, Media & Design, Bouve College of Health Sciences, Northeastern University, Boston, MA 02115, USA; cvsousa89@gmail.com; 2Genomic Science and Biotechnology, Catholic University of Brasilia, Brasilia 70790-160, Brazil; rinaldo.pereira@catolica.edu.br; 3Institute of Primary Care, University of Zurich, 8091 Zurich, Switzerland; thomas.rosemann@usz.ch; 4Exercise Physiology Laboratory, 18450 Nikaia, Greece; pademil@hotmail.com; 5Medbase St. Gallen Am Vadianplatz, 9001 St. Gallen, Switzerland

**Keywords:** swimming, cycling, running, ultra-endurance, recovery

## Abstract

The present case study analyzed performance, pacing, and potential predictors in a self-paced world record attempt of a professional triathlete to finish 40 Ironman-distance triathlons within 40 days. Split times (i.e., swimming, cycling, running) and overall times, body weight, daily highest temperature, wind speed, energy expenditure, mean heart rate, and sleeping time were recorded. Non-linear regressions were applied to investigate changes in split and overall times across days. Multivariate regression analyses were performed to test which variables showed the greatest influence on the dependent variables cycling, running and overall time. The athlete completed the 40×Ironman distances in a total time of 444:22 h:min. He spent 50:26 h:min in swimming, 245:37 h:min in cycling, 137:17 h:min in running and 11:02 h:min in transition times. Swimming and cycling times became slower across days, whereas running times got faster until the 20th day and, thereafter, became slower until the 40th day. Overall times got slower until the 15th day, became faster to 31st, and started then to get slower until the end. Wind speed, previous day’s race time and average heart race during cycling were significant independent variables influencing cycling time. Body weight and average heart rate during running were significant independent variables influencing running performance. Cycling performance, running performance, and body weight were significant independent variables influencing overall time. In summary, running time was influenced by body weight, cycling by wind speed, and overall time by both running and cycling performances.

## 1. Introduction

Triathlon races of the classical Ironman distance (i.e., 3.8 km swimming, 180 km cycling, and 42.195 km running) [1] and ultra-triathlon races with multiple times the Ironman distance such as Double Iron ultra-triathlon (i.e., 7.6 km swimming, 360 km cycling, and 84.4 km running), Triple Iron ultra-triathlon (i.e., 11.4 km swimming, 540 km cycling and 126.6km running), Quintuple Iron ultra-triathlon (i.e., 19 km swimming, 900km cycling, and 221 km running) and Deca Iron ultra-triathlon (i.e., 38 km swimming, 1800 km cycling, and 422 km running) are of increasing popularity [2,3].

Pacing during endurance and ultra-endurance performance is very important for a successful race outcome. Different pacing strategies are known such as: negative pacing (i.e., the athlete becomes faster during the performance); all-out pacing (i.e., limited to extremely short performances of ≤30 s); positive pacing (i.e., the athlete becomes slower during the performance); even pacing (i.e., the performance is constant over time); parabolic-shaped pacing (i.e., speed increases and decreases); and variable pacing (i.e., change of negative and positive pacing) strategies [4]. 

In longer triathlon races, pacing has been investigated for both the cycling and running split in elite female and male Ironman triathletes, where both women and men adopted a positive pacing during both cycling and running splits [5]. In recent decades, the length of the triathlon race distances has increased. While distances of 2× to 10× Ironman are regularly held, only one official race of 30×Ironman-distance triathlons was ever held in autumn 2013 [6]. To date, only one athlete went beyond the 30 days in a self-paced event completing 33 Ironman-distance triathlons in 33 days also held in summer 2013 [7].

Ultra-endurance athletes are pushing their limits to go for the ultimate limit in endurance performance [8]. In ultra-running, the limit has most probably achieved with crossing a continent [9]. In the ‘Trans Europe Foot Race 2009’, a small sample of a few dozens of ultra-marathoners covered the distance of 4487 km from South Italy to North Cape [10]. However, the total distance of 4487 km within 64 days [11] was outperformed in 2016 by the French ultra-marathoner Patrick Maladin with 10,000 km in 100 days (99 days 4 h 12 min) with the daily performance of a 100 km ultra-marathon [12]. However, in 2019, he broke his own record while he crossed the United States of America from New York to Los Angeles (4801 km) within 46 days and Canada from Vancouver to Halifax (5931 km) within 56 days to complete the overall distance of 10,732 km within 102 days 18 h and 48 min [13].

Triathletes push their limits as much as the ultra-runners with completing daily an Ironman-distance triathlon for as many days as possible. While the longest scientifically verified self-paced event was 33 Ironman-distance triathlons in 33 days in summer 2013 [7], we present here the pacing in the first and only athlete to complete a scientifically verified (all data collected and open available for detailed analysis) self-paced world record attempt to finish 40 Ironman-distance triathlons within 40 days held in autumn 2019. We also investigated potential predictor variables for split disciplines (i.e., cycling and running) and overall performance.

## 2. Materials and Methods 

### 2.1. Ethical Approval

This study was approved by the Institutional Review Board of Kanton St. Gallen, Switzerland, with a waiver of the requirement for informed consent of the participant as the study involved the analysis of publicly available data. Heart rates [14] and body weight [15] are free available from the blog of the athlete. The study was conducted in accordance with recognized ethical standards according to the Declaration of Helsinki adopted in 1964 and revised in 2013. 

### 2.2. Athlete

The athlete (36 years old, body mass 74 kg, body height 1.78 m, body fat percentage 7.3%) is a professional world class level ultra-endurance athlete competing in ultra-marathon running and ultra-triathlon races. Until the start of the event, he had finished 17 official ultra-triathlon races. These were 7 Double Iron ultra-triathlons, 4 Triple Iron ultra-triathlons, 1 Quadruple Iron ultra-triathlon (4× Ironman in 4 days), 1 Quintuple Iron ultra-triathlon, 1 Deca Iron ultra-triathlon as 10× Ironman in 10 days and 1 Deca Iron triathlon with 38 km swimming, 1800 km cycling and 422 km running. Apart from the official races, he finished 2 self-paced ultra-triathlons (5× Ironman-distance in 5 days and 20× Ironman-distance in 20 days). The personal best times for the official races are: 19:42:57 h:min:s (Double Iron ultra-triathlon—official world record June 2019), 32:57:01 h:min:s (Triple Iron ultra-triathlon), 53:08:27 h:min:s (4×Ironman-distance triathlons in 4 days), 72:55:29 h:min:s (Quintuple Iron ultra-triathlon), 234:31:15 h:min:s (Deca Iron ultra-triathlon), and 108:48:58 h:min:s (10× Ironman-distance triathlons in 10 days). He has won 5 races and come to second place at 7 times. Most of his personal best times were the second fastest times ever achieved behind the world record. His personal best time in a single Ironman is 9:50:30 h:min:s in 2013.

Apart from the ultra-triathlons, he competed in 13 ultra-marathons. He has completed ‘TransGranCanaria’ ultra-run three times in 2015, 2016 and 2018 (i.e., 125 km, 7500 m elevation gain, best time 16:53:55 h:min:sin 2018), ‘Vaarojen Marathon’ in 2011 (86 km, 2325 m elevation gain), ‘Axtrail Ultra Trail Aldeias Do Xisto’ in 2012—Utax (82 km, 5000 m elevation gain), ‘The North Face Ultra-Trail Du Mont-Blanc^®^’ in 2013 - Tds^®^ (119 km, 7250 m elevation gain), ‘Lanzarote Ultra-Trail’ (84 km, 2674 m elevation gain), ‘Haanja Ultra100’ (100 km, 1125 m elevation gain) in 2014, 2015 and 2017—he won all those three races, with the best time 8:03:48 h:min:s.

In the last three years before the planned event of 40 Ironman-distance triathlons in 40 days, his training consisted primarily of swimming, cycling, running, and body strength workouts. On average, he invested annually ~1100 h of training with ~900 km of swimming, ~12,000 km of cycling and ~5000 km of running. He developed his own training principle where his training units have short transitions. With this principle, he saves time and trains the transition from one discipline to the other. As an example, he starts his training day with running 11 km, changes to swimming 4.4 km, changes then to running 13 km, followed by cycling for 1.5 h. He likes to split the distance. In order to run 84.4 km (two marathons) in a Double Iron ultra-triathlon in 6 h, he needs to run long distances, but by splitting. As an example, when running 64.1 km, he starts with running 16.4 km, after a break of 1 h, he runs 15.4 km, after a small break he runs 20.3 km, and after a further short break 12 km. With this principle, he can run the same distance a few days later in three (21.7 km, 25.9 km, and 16.3 km) compared to four runs.

### 2.3. Event, Equipment, Support, and Measurements

The concept of the event was a self-paced world record attempt to achieve as the first athlete in the world 40 Ironman-distance triathlons in 40 days. The event was held in Fuerteventura (Las Playitas, Canary Island). The challenge started 28th of September, 2019. The athlete followed no specific pacing strategy. Every day of the event, he tried to swim, cycle and run as fast as possible with the consideration that he must be able to compete the following day in the same manner. He always selected the specific speed in all three disciplines according to his subjective feelings without monitoring of heart rate or split times. He tried to get through the transition zones and food points as fast as possible. The athlete is a very experienced competitive athlete to listen to body signals and tries to prevent potential problems such as overheating, dehydration, lack of energy, muscle overload, muscle stiffness, general fatigue, or lack of sleep. Based upon his previous experience he knows himself very well, and most of the time he can notice and understand these signals. He monitored his body weight daily (every morning) using Garmin Index Smart Scale [16]. The energy expenditure was measured using the heart rate method with Garmin Forerunner 935 [16]. 

The swim took place in the 50-m outdoor pool at Playitas resort [17] where the water temperature was ~26–27 °C. He used every day a sleeveless wetsuit [18]. The bike course was one loop of 180 km with an elevation of 2079 m (measured by Google maps and Garmin Forerunner 935). The bike loop was open to car traffic. He used a Trek^®^ Concept 7.5 triathlon bike with regular Bontrager^®^ race wheels [19]. The run took place at Playitas resort with 12.5 laps of 3.375 km each (measured by Google maps and Garmin Forerunner 935). He started from the highest point of the lap and had to then complete 12 full laps. The last half lap consisted in the descent to sea level. He measured the elevation by running two full laps and divided the whole elevation (54 m) by two resulting in 27 m elevation difference per running lap. Running shoes were mostly different types of Saucony^®^ shoes (i.e., Kinvara 9, Kinvara 10, Everrun, Type A8, Fastwitch 8 and Fastwitch 9) [20]. One day, he used also Hoka^®^ One Tracer’s shoes [21] but changed them during the run because he felt uncomfortable.

The athlete started the swim for Days 1–29 at 07:40 a.m. After turning the clock one hour back at the end of October, he started at 07:30 a.m. This procedure was chosen in order to be able to run as long as possible in daylight due to the change from summer to winter time. The weather was very windy, mostly sunny and basically without rain. Two times was some rain. The average high temperature in October in Fuerteventura is ~26 °C, and no more than two days with little rain can be expected [22]. Daily temperatures (i.e., the daily highest air temperatures) were obtained from a local weather report [23]. All times (i.e., swimming, cycling, running, transition, overall, and sleeping time) were measured using a Garmin^®^ watch (Forerunner 935) and by the support crew using a stop watch CHRO301. Official times were those times measured and recorded by the support crew.

During the 40 days he had a team of 25 persons. They worked in shifts and were responsible for timing, nutrition, and material. In swimming, he had always two persons where one was counting the laps. During the swim, he did not eat anything, only had a bit of a sports drink. After the swimming, he ate one big bowl of cereal or rice porridge with cinnamon and cactus jam. Then, the crew changed where two new persons accompanied him by car on the bike. During the 180 km, he had six food stations where he made a short stop for food. In case of very hot weather, 3–4 additional aid stations were added as drink stations where he passed without stopping. 

During cycling, he drank mostly water with minerals and vitamins [24], sports drinks, iced tea, sometimes one cup a coffee, pure 100% orange juice, non-alcoholic beer and lots of pure water. On the bike course, he always consumed salt tablets (www.enervit.com). Between the food stations, he consumed lots of gels. At the food stations, he consumed croissants with cheese, lots of muesli, fruits (i.e., watermelon, kiwi, melon, grapes), fried eggs, pancakes with Nutella^®^, very many different sweets (i.e., chocolate, biscuits, Enervit^®^ protein bars). At the last food station and also right after finish of the cycling, he ate a bowl of Kellogg’s^®^ cereals with orange juice.

During the run, he had two persons as support crew where one was following him by bike and supported him with food and drinks and the other one was located at the food station to prepare nutrition. On the run course, he stopped only for comfort breaks and changes of shirts or running shoes. During the run, he consumed Coca Cola^®^ (about 1,4 l/day), iced tea (about 0.7–0.8 l/day), lots of watermelon and some biscuits. During the run, he always consumed salt tablets [25]. Overall, during the 40 days, he consumed about 700 salt tablets.

After the run, he always had a bowl of Kellogg’s^®^ and often ate lots of sweets. Then 30 min later, he had dinner at Playitas restaurant (Swedish table). He ate pasta, rice, fries with different sauces, bread with cheese and jam, and lots of different homemade cakes. An hour later, he had a massage of about one hour. Before and after the massage, he ate bread with egg butter, omelette with cheese, sweets, nuts and raisins and/or crispbread with peanut butter and he drank lots of iced tea and mineral water. During the night, he ate food he brought with him from the restaurant (e.g., pasta, cakes etc.). In the morning, he drank coffee, ate sweets and Kellogg’s^®^, and sometimes he took one RedBull^®^ before swimming. He was sleeping in an apartment of Playitas resort. The sleeping time was measured using Garmin^®^ Forerunner 935.

During the challenge, he faced many different problems. At the beginning, he had problems with weight loss because he was not used to expending so much energy per day. Body mass loss was too fast in the first days and the loss continued without stopping toward the half of the event. In the first days, he had also problems with the hot weather (temperature was over 30 °C) and on the 6th day he felt overheated at the end of the bike. He took 25 min in T2 zone to recover using ice and consuming cold drinks. At the beginning of the challenge, he had problems with the bike position since he was not used to sit on the saddle for such a long time. After six days he got used to it. During the challenge, he had only three blisters. To avoid blisters, he used finger socks and Vaseline. The mentally hardest days (self-reported by the athlete and not utilizing a standard measurement) were days 6, 12 and 18 and were mainly caused by very difficult wind conditions (i.e., strength or direction). He was physically in good conditions and had no muscular problems, However, these three days, it took him a lot of willpower to cope with the strong winds while cycling. By the end of these days, he felt mentally tired (self-reported by the athlete and not utilizing a standard measurement).

The second half of the challenge, he had problems with muscles that caused some discomfort during the run. During the 40 days, he suffered from different muscular problems, but no overuse injuries. The athlete considered the second half of the challenge mentally and physically easier than the first half. Mentally because after passing day 20, he was doing something new since his previous challenge was 20 Ironman-distance triathlons in 20 days. So, every step forward was pushing his limits farther. The first part was harder because at the beginning he was not used to it and he got under great stress. Also, he needed some time to get used to the hot weather. When some days the temperature dropped a bit, he started to feel a bit chilly. The harshness of the whole challenge was hidden in the weather conditions (i.e., wind) with the direction and/or the strength of the wind. At first, both body and mind needed time to adjust to the wind. As the days went on, he got mentally stronger. He believed that when he had passed the previous point (20 days), it would become mentally easier and he could compete faster.

### 2.4. Statistical Analysis

Exploratory analysis of the data was performed, reporting mean, standard deviation, minimum, maximum, and coefficient of variance (CV) of the variables. A paired t-test was applied to compare average heart rate of cycling and running. Repeated measures ANOVA was applied to test the time effect over performance with the average of every 10 race days. Non-linear regressions (2nd and 3rd order polynomial) were applied to investigate changes in split (swimming, cycling, and running) and overall times. Multivariate regression analyses were performed to test which variables had the greatest influence on the dependent variables. We ran three models: model 1 = cycling performance as the dependent variable, model 2 = running performance as the dependent variable, and model 3 = overall performance as the dependent variable. All procedures were conducted using Statistical Software for the Social Sciences (IMB^®^ SPSS v25. Chicago, Il, USA) and GraphPad Prism (GraphPad Prism v8. San Francisco, CA, USA). The statistical significance was set at *p* ≤ 0.05.

## 3. Results

The athlete completed the 40x Ironman distances in a total time of 444:22 h:min. He invested 50:26 h:min in swimming, 245:37 h:min in cycling, 137:17 h:min in running and 11:02 h:min in transition times. Performance in swimming and cycling became reduced over the days, whereas running performance improved until the 20th day and became reduced to the 40th day (Figure 1). Second-order non-linear regressions showed the best fit for swimming, but low or non-fitting models for cycling, running and overall race time. The coefficients of variance (CV) were higher in running, followed by cycling and swimming (Table 1). Regarding overall times, the last two days were the slowest ones, which were also the slowest regarding running and cycling performance. A third-order polynomial non-linear regression was the best fit for overall performance (R^2^ = 0.732). Ten-day race time average showed a significant time effect for swimming (F = 17.6; *p* < 0.001), running (F = 7.5; *p* = 0.011) and overall performance (F = 4.2; *p* = 0.034), but no for cycling (Figure 2).

Physiological variables are shown in Table 2. Times of the event increased (got slower) until day 15, improved to day 31 and then again increased until the end indicating a sinusoidal pattern (Figure 3A). Average heart rate was significantly higher in running in comparison to cycling (t_39_ = 21.7; *p* < 0.001) (Figure 3B). Morning body weight reached the highest peak on the 6th day with 77.8 kg and the lowest peak on the 23rd and 36th day with 73.6 kg (Figure 2C). Wind speed and temperature showed erratic behaviors throughout the days (Figure 3D), with wind speed going up to 52.5 km/hon the 35th day, and the temperature reaching 35 °C on the 3rd day. Energy expenditure was higher in the first 10 days, and then varying from ~7500 to 8000 kcal/day in the following days (Figure 3E). Finally, with the exception of the first night with only 232 min of sleep, sleep duration varied between ~370 and ~480 min/night throughout the race days (Figure 3F). 

Table 3 presents the results of the four multivariate regression models. The first multivariate regression model (swimming model) used swimming performance as the dependent variable and showed statistical significance (*p* = 0.010) with an adjusted R^2^ of 0.28. The only significant independent variables to influence swimming performance was wind speed (*p* = 0.005), and a trend (*p* = 0.062) for body weight. The second multivariate regression model (cycling model) used cycling performance as the dependent variable and showed statistical significance (*p* < 0.001) with an adjusted R^2^ of 0.78. Wind speed, overall time in the previous day and average heart race during cycling were the significant independent variables (*p* < 0.05) to influence cycling performance. The third multivariate regression model (running model) used running time as the dependent variable and showed statistical significance (*p* < 0.001) with an adjusted R^2^ of 0.69. Body weight and average heart rate during running were the significant independent variables (*p* < 0.05) to influence running performance. The fourth multivariate regression model (overall model) used overall time as the dependent variable and showed statistical significance (*p* < 0.001) with an adjusted R^2^ of 0.98. Cycling performance, running performance and body weight were the significant independent variables (*p* < 0.05) to influence overall time.

## 4. Discussion

In this self-paced world record attempt, we found that (i) performances in cycling and running were predictive for overall time, (ii) overall time was predictive for cycling performance of the following day, (iii) wind speed was predictive for swimming and cycling time, (iv) body mass was predictive for running and overall time, (v) heart rate was predictive for cycling and running time, (vi) temperature and sleep time were not predictive to performance, and (vii) the highest variation occurred in the marathon split times.

### 4.1. Cycling and Running were Predictive for Overall Race Time, But not Swimming

An important finding was that both cycling and running times were predictive for overall race time. While the cycling split is highly predictive in a single Ironman-distance race [26,27], both cycling and running are predictive in longer triathlon distances [28,29]. The contribution of swimming to overall performance in an ultra-triathlon is lower than for cycling and running [30]. A study investigating the performance level and race distance on pacing in ultra-triathlons (i.e., Double, Triple, Quintuple and Deca Iron ultra-triathlon) showed that the fastest ultra-triathletes spent relatively more time in swimming and cycling and less time in running, highlighting the importance of the role of the latter discipline for the overall ultra-triathlon performance [28]. Running seems, however, also important in a single Ironman-distance race. When split times (i.e., swimming, cycling, and running) and overall race times of 343,345 athletes competing between 2002 and 2015 in 253 different Ironman-distance triathlon races were analyzed, it was shown that the fastest Ironman-distance triathletes were the relatively fastest in running and transition times [31]. Similarly, a study investigating Triple Iron ultra-triathletes showed that running rather than cycling performance seemed to be the most important predictor for overall race time [32]. Overall, both cycling and running seemed of importance in ultra-endurance multi-stage triathlon races, but not swimming.

### 4.2. The Aspect of Experience and Age

The present athlete was able to perform during 40 days an Ironman-distance triathlon with an average race time of ~11 h. per day with 10:28 h:min for the fastest and 12:00 h:min for the slowest day. His personal best time in Double Iron ultra-triathlon was in June 2019 an official world record and his personal best time in 10× Ironman in 10 days was the second fastest time in history. His personal best time in a single Ironman is 9:50:30 h:min:s in 2013. Since then, he never competed again in a single Ironman triathlon.

It is well known that previous experience is important for an ultra-endurance performance such as an Ironman [33] or an ultra-triathlon [34,35,36,37]. When all female and male ultra-triathletes who had finished at least one Double Iron ultra-triathlon, one Triple Iron ultra-triathlon, one Quintuple Iron ultra-triathlon, and one Deca Iron ultra-triathlon, it was shown that fast race times in shorter ultra-triathlon races (i.e., Double and Triple Iron ultra-triathlon) were more important than a large of number finished races in order to achieve a fast race time in a longer ultra-triathlon (i.e., Quintuple and Deca Iron ultra-triathlon) [34]. In successful finishers in a Deca Iron ultra-triathlon, both the number of finished Triple Iron ultra-triathlons and the personal best time in a Triple Iron ultra-triathlon were related to overall race time [36]. Furthermore, the athlete is at the best age (36 years) for a fast Ironman triathlon performance since the most competitive age for male athletes is ~35 years for a top performance in an Ironman triathlon [38,39].

### 4.3. Sinusoidal Change in Overall Race Time 

A further finding was the sinusoidal change in overall time with a decrease in performance until the 15th day, an improvement in performance up to the 31st day, and a decrease until the end of the event. To date, only one study investigated the pacing in a multi-stage triathlon longer than a Deca Iron ultra-triathlon. In the first and only Triple Deca Iron ultra-triathlon held until now, the daily performance remained unchanged across the 30 days (i.e., even pacing) in the eight male finishers [6]. Different intrinsic and extrinsic factors such as motivation, weather, course, nutrition etc. [40,41,42] might explain the difference between the finishers in a Triple Deca Iron ultra-triathlon and the athlete in the present case study.

### 4.4. The Aspect of Heart Rate and Energy Expenditure

Average heart rate was significantly higher in running in comparison to cycling splits, and heart rate during the split performances was predictive for the performance in the corresponding split. It is well known from laboratory studies with triathletes that heart rate and energy expenditure is higher in running compared to cycling [43]. While the measurement of heart rate with a wrist-worn device might be reliable [44], the reported energy expenditure from these devices should be interpreted with caution [45], given their potential bias and error [46], the current wrist-worn activity trackers are most likely not accurate enough [26]. However, these devices might be suitable for the use in interventions of behavior change as they provide feedback to user on trends in energy expenditure [46]. Energy expenditure reported by wrist-worn devices differs between different sports disciplines (e.g., walking and running) with a general increase as exercise intensity increased [46]. Interestingly, heart race correlated with cycling and running performance, but not the measured energy expenditure with the wrist-worn device. However, this is the only way to get a note about the energy expenditure under such conditions.

### 4.5. Energy Expenditure, Loss in Body Mass and Body Weight as Predictor in Performance

Mean energy expenditure per Ironman-distance was ~7780 kcal and the athlete lost a total of ~4 kg of body mass. Although the determination of energy expenditure using a wrist-worn device might be questionable, the mean energy expenditure per Ironman was practically the same as has been reported for an athlete in a 10× Ironman with 7544 ± 913 kcal per day when using a portable heart rate monitor [47]. For a single Ironman triathlon, energy expenditure is, however, higher at ~10,036 ± 931 kcal [48] to ~11,009 ± 664 kcal [49]. The difference might be explained that intensity during a single Ironman triathlon is considerably higher than when finishing every day an Ironman-distance triathlon during multiple days. This is evidenced due to this specific athlete’s daily performance as compared to their personal best in a single Ironman-distance event.

A loss in body mass in a single Ironman triathlon [50,51] and an ultra-triathlon [52,53,54,55] is a common finding where the loss in body mass in an Ironman triathlon is related to a loss in skeletal muscle mass [50]. This loss in muscle mass is due to depletion in muscle glycogen stores [50]. In ultra-endurance triathletes competing in longer races than an Ironman triathlon, mainly body fat is lost. The loss in body mass in a Deca Iron ultra-triathlon was due to a loss in body fat [54] and in Triple Iron ultra-triathletes the loss in body mass was related to the loss in body fat [55]. While the loss in body mass in a single Ironman triathlon is ~2 kg for male athletes [50,51], the present athlete lost only ~4 kg during 40 days. Obviously, he was well experienced to preserve his body mass during these 40 days. However, the absolute loss in solid body mass might have been higher since total body water increases in a multi-stage triathlon such as a Deca Iron ultra-triathlon [47,56]. In a Deca Iron ultra-triathlon, both body mass and fat mass decrease [36,47] whereas lean body mass [36] and total body water increase [47]. The increase in lean body mass is due to the increase in total body water [57]. While multi-stage ultra-triathletes seem to become overhydrated, finishers in a single Ironman are hypo- to dehydrated [58]. Overall, multi-stage ultra-triathletes seem able to preserve their body mass. In a Deca Iron ultra-triathlete, the overall loss in body mass was ~1 kg although the energy deficit was ~11,480 kcal during the 10 days [47]. 

Body mass was predictive for running and overall time. For a single Ironman triathlon, body mass as an anthropometric variable is not predictive for race time, but rather previous experience such as the personal best time in an Olympic distance triathlon and in a marathon [59]. In the present athlete, body mass decreased through the event and we might assume that the loss in body mass was ‘ergogenic’. Studies investigating marathoners [42] and 100 km ultra-marathoners [60] showed that a loss in body mass was related to faster race times. A study investigating marathon runners showed a relationship between running speed during training and percent body fat [61]. 

### 4.6. The Influence of Environmental Factors (e.g., Wind Speed and Temperature) and Recovery

It was shown that wind speed was predictive for swimming and cycling, but the daily highest temperature was not related to performance. It is well known that environmental conditions such a wind and heat have an influence on ultra-endurance performance [62]. While it is known that wind has an influence on ultra-cycling performance [63], one might assume that also heat might impair ultra-endurance performance [64]. Triathletes competing under thermal stress conditions in the ‘Ironman Hawaii’ reached a state of hyperthermia during the marathon [65].It was observed that overall time was predictive for cycling performance of the following day and sleep time was not related to performance. It seems that the athlete was able to find his best way of intensity and recovery since it is well known that sleep deprivation would lead to an impaired athletic performance [66].

### 4.7. Limitations

Although this case study provides a lot of physiological and technical details, some limitations must be acknowledged. The daily measurement of body composition using a bioelectrical impedance analysis or radiological procedures like BIA [67], DEXA [68], or MRI [69] would help monitor changes in body composition. The inclusion of the last two days with very slow overall times (outlier) might have had an influence on the analysis of the data. Missing data for humidity [70], wind direction [71] and nutrition [72] also might have had an influence on the outcome of the analysis. We only used the daily maximum of air temperature and wind speed which might have had an influence on our analysis.

### 4.8. Practical Applications

This case study shows that it is possible to perform an Ironman-distance triathlon daily for 40 days. The findings in this study may help any athlete intending to outperform the performance with a faster time per Ironman or to finish more consecutive Ironman-distance triathlons. 

## 5. Conclusions

In summary, a fast running performance is significantly influenced by a low body weight, a fast cycling performance is significantly influenced by both a low wind speed and by the previous day’s effort, and a fast overall performance is significantly influenced by both fast running and fast cycling performances. This study could be of assistance for coaches and athletes preparing for similar challenges. More in-depth physiological responses (i.e., body composition measures, sleep quality, inflammation, oxidative stress) to such a challenge with or without a follow-up could be the next step. A highly trained professional triathlete with extensive previous experience is able to finish daily an Ironman-distance triathlon on consecutive 40 days where cycling and running performance are highly predictive of overall time. 

## Figures and Tables

**Figure 1 ijerph-17-02390-f001:**
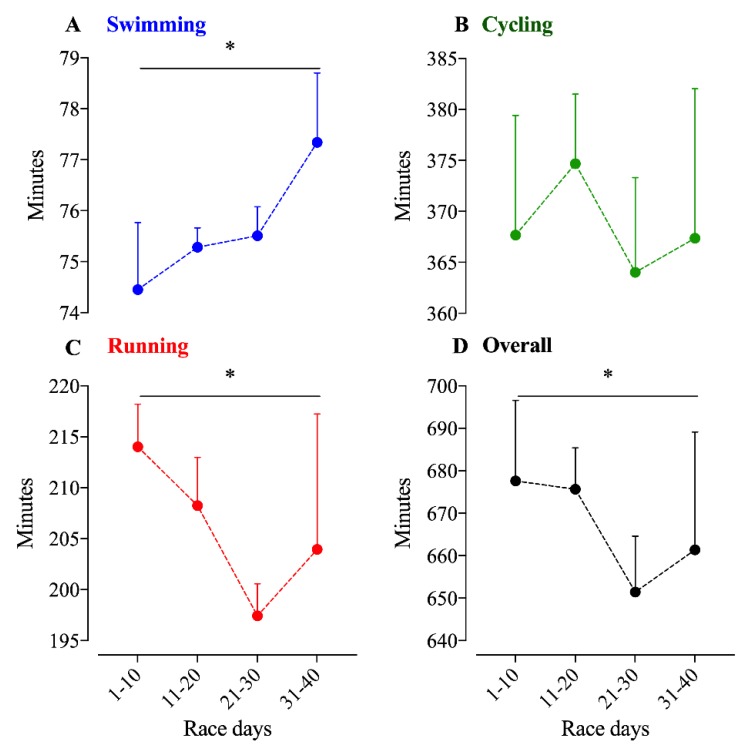
Ten-day average times of swimming, cycling, running, and overall time in 40 Ironman-distance triathlons. *: statistical significance (*p* < 0.05) for time-effect. (**A**): Swimming time; (**B**): Cycling time; (**C**): Running time; (**D**): Overall time.

**Figure 2 ijerph-17-02390-f002:**
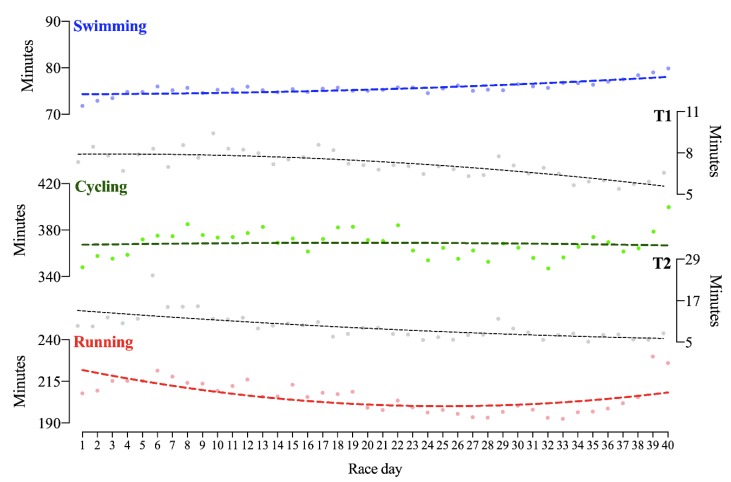
Times of swimming, cycling, running, and transition in 40 Ironman-distance triathlons in 40 days.

**Figure 3 ijerph-17-02390-f003:**
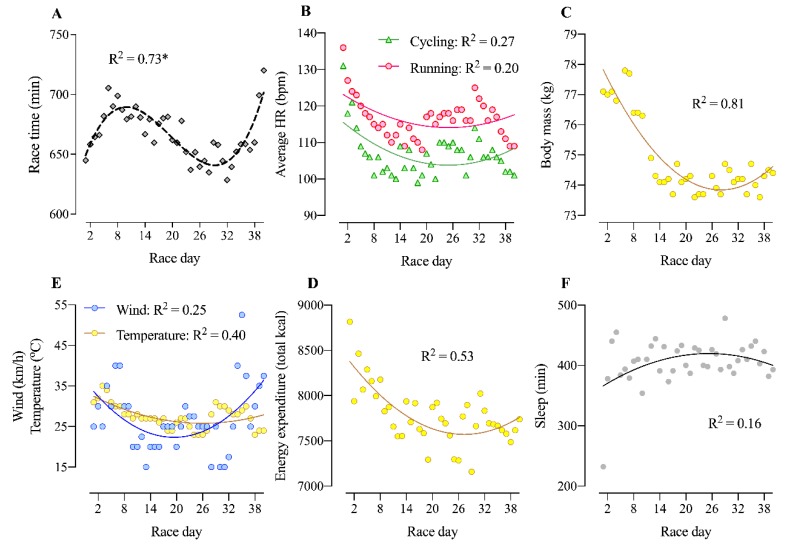
Non-linear regression for overall performance, weather and physiological variables in 40 Ironman-distance triathlons in 40 days. *: third-order polynomial fit; all other regressions were second-order polynomial fits. (**A**): overall performance; (**B**): average heart rate (HR) during cycling and running; (**C**): body mass; (**D**): wind speed and average temperature; (**E**): energy expenditure; (**F**): sleep minutes between race days.

**Table 1 ijerph-17-02390-t001:** Split and overall times (minutes), coefficient of variance (CV) and goodness of fit in 2nd order polynomial regression (R^2^) for 40 Ironman-distance triathlons for 40 consecutive days.

Times	Mean (±SD)	Min–Max	CV	R^2^
Swimming split time	75.6 (1.4)	71.8–79.9	1.9	0.63
Cycling split time	368.4 (11.3)	347.0–399.8	3.1	0.00
Running split time	205.9 (9.5)	192.6–230.0	4.6	0.41
Overall time	666.5 (21.0)	628.5–720.0	3.1	0.13

**Table 2 ijerph-17-02390-t002:** Total energy expenditure, average heart rate, sleep time and body weight during 40 Ironman-distance triathlons for 40 consecutive days.

Measured Variables	Mean (±SD)	Min–Max
Energy expenditure (total kcal)	7780.2 (319.5)	7159.0–8815.0
Average heart rate in cycling (bpm)	106.9 (6.3)	99.0–131.0
Average heart rate in running (bpm)	116.5 (5.6)	108.0–136.0
Sleep duration (min)	405.5 (37.6)	232.0–478.0
Morning body weight (kg)	74.8 (1.3)	73.6–77.8

**Table 3 ijerph-17-02390-t003:** Results of the multivariate regression models to test which variable has the highest influence in swimming, cycling, running and overall time in 40 Ironman-distance triathlons in 40 days.

	Independent Variables	Standardized β	*p*-Value
Model 1 Swimming	Sleep time	−0.09	0.59
	Body weight	−0.46	0.06
	Wind speed	0.47	0.01
	Overall time previous day	0.10	0.57
	Temperature	−0.18	0.40
Model 2 Cycling	Swimming time	−0.01	0.96
	Sleep time	0.04	0.68
	Body weight	−0.08	0.57
	Wind speed	0.29	0.01
	Overall time previous day	0.51	<0.01
	Temperature	0.12	0.33
	Average heart rate	−0.61	<0.01
Model 3 Running	Swimming time	0.221	0.12
	Cycling time	0.11	0.53
	Sleep time	−0.13	0.25
	Body weight	0.64	<0.01
	Wind speed	−0.01	0.97
	Overall time previous day	0.08	0.61
	Temperature	0.19	0.25
	Average heart rate	−0.50	0.02
Model 4 Overall	Swimming time	0.04	0.29
	Cycling time	0.60	<0.01
	Running time	0.45	<0.01
	Sleep time	−0.01	0.71
	Body weight	0.14	0.01
	Wind speed	−0.05	0.14
	Overall time previous day	0.02	0.69
	Energy expenditure	0.00	0.99
	Temperature	0.02	0.59

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
