# Peer review of "Self-Selected Pacing During a World Record Attempt in 40 Ironman-Distance Triathlons in 40 Days"

_ijerph, 2020, doi:10.3390/ijerph17072390_

Round 1

Reviewer 1 Report

- The introduction is quite weak. It should be improved, presenting the relevance of the study and with a more relevant literature review. It is not clear “why” the study was done and what should be expected with its results

- The statistical analysis could be improved. For example: why did authors not test for differences in the performance across the days (in the overall performance, as well as in each of the splits)?

- What is the news of the results? Most of them are not yet expected? How these results can be useful for other athletes/coaches, if they reflect the performance/adaptability of one single athlete?

- Some typos must be corrected (authors should carefully read and correct the manuscript)

Author Response

Reviewer 1

The introduction is quite weak. It should be improved, presenting the relevance of the study and with a more relevant literature review. It is not clear “why” the study was done and what should be expected with its results

Answer: We agree with the expert reviewer and expanded the Introduction by adding and/or changing the text with: Ultra-endurance athletes are pushing their limits to go for the ultimate limit in endurance performance [Adv Exp Med Biol. 2007;618:255-76]. In ultra-running, the limit has most probably achieved with crossing a continent [BMC Med. 2012 Jul 19;10:76]. In the ‘Trans Europe Foot Race 2009’, a small sample of a few dozens of ultra-marathoners covered the distance of 4,487 km from South Italy to North Cape [BMC Med. 2012 Jul 19;10:77]. However, the total distance of 4,487 km within 64 days [J Sports Med Phys Fitness. 2019 Oct;59(10):1608-1621] was outperformed by the French ultra-marathoner Patrick Maladin with 10,000 km in 100 days (99 days 4 hours 12 minutes) in 2016 with daily a 100 km ultra-marathon [http://patrickmalandain-ultrarun.com/parcours-10000km]. However, in 2019, he broke his own record while he crossed the United States of America from New York to Los Angeles (4,801 km) within 46 days and Canada from Vancouver to Halifax (5,931 km) within 56 days to complete the overall distance of 10,732 km within 102 days 18 hours and 48 minutes [http://patrickmalandain-ultrarun.com/#us]. Similarly, triathletes seem to push also their limits with completing daily an Ironman-distance triathlon for as many days as possible. While the longest scientifically verified self-paced event was 33 Ironman-distance triathlons in 33 days in summer 2013 [7], we present here the pacing in the first and only athlete to complete a scientifically verified self-paced world record attempt to finish 40 Ironman-distance triathlons within 40 days held in autumn 2019. We also investigated potential predictor variables for split disciplines (i.e. cycling and running) and overall performance.

The statistical analysis could be improved. For example: why did authors not test for differences in the performance across the days (in the overall performance, as well as in each of the splits)?

Answer: The athlete performed an Ironman distance triathlon in each day, for 40 days. Thus, performance data is available for each day. Since the athlete do not have any race time exactly the same to the other it is statistically correct to assume that each day were different from the other. There is no statistical test to compare a single value vs. single value, t-tests consider degrees of freedom and ANOVAs consider variance to compare groups of data, not single data. However, we averaged every 10 days and compared with a repeated measures ANOVA, including more results and a new figure. Please check Figure 2 and changes in text highlighted in red.

What is the news of the results? Most of them are not yet expected? How these results can be useful for other athletes/coaches, if they reflect the performance/adaptability of one single athlete?

Answer: We agree with the expert reviewer and add a section ‘practical applications’ with: This case study shows that it is possible to perform an Ironman triathlon daily for 40 days. The findings in this study may help any athlete intending to outperform the performance with a faster time per Ironman or to finish more Ironman triathlons.

Some typos must be corrected (authors should carefully read and correct the manuscript)

Answer: The manuscript was checked again for English and improved where necessary.

Reviewer 2 Report

The authors investigate an interesting and novel case-study of a repetitive ultra-endurance performance.  The case presented includes new findings related to pacing, recovery, and impact of physiology and environmental conditions on repeat performance of an Ironman-distance triathlon. We applaud the authors on their ability to compile data from this 40-day effort.

There are several serious concerns that preclude me from recommending this study for publication in its current state.  There are omissions in the methods, discussion and conclusion and a lack of identification of any study limitations. Additionally, significant revision for English language and style is required.  The manuscript as currently written is difficult to read and uses generic adjectives instead of specific language when referring to important segments of the study.

The term Ironman triathlon is a corporate brand.  Therefore as this event was not sanctioned or organized by the WTC Ironman corporation, all instances should be referred to as "Ironman-distance" events throughout the manuscript.

Ethically, in section 2.1, the study states that all data was publicly available.  If so, please include the direct link.  I am concerned that data related to heart rate and body weight was however not publicly available and therefore this does not meet the standard for a waiver.

In describing the athlete in section 2.2 a statement is made that all training was "short and fast."  Please clarify this generic description.  Without qualifiers it is difficult to know what short and fast mean.  Please define. Include an average training week for this athlete.

In section 2.3 there are conflicting statements regarding whether HR was available or not.  Line 99 indicates the athlete did not check HR but line 102 indicates that HR and EE were monitored.  Please clarify.

Add an indication of elevation changes for the run course.

A better definition of ambient environmental conditions is needed.  "very windy, mostly sunny and basically without rain" is an obtuse categorization of the weather.  What were the means and ranges for these conditions?  What was the direction of the wind relative to the bike and run course? Consider adding a table. Additionally it would be useful to consider relative humidity in the context of it's effects on performance.  The location of the event precludes generalization and so specific details of the conditions must be better described.

While the nutrition section gives an idea of the caloric and macro-nutrient intake of the athlete, term such as "big bowl", "lots of gels", "Kellogg's cereal" are too generic for a single study case study.  What were the quantities involved and the timing of the intake? Line 144 indicates a numeric value but there are no units indicated.  Please correct.  The paragraph on running nutrition also includes a statement on salt intake during the bike (line 145) and a statement of morning intake (line 153).  Please remove from this section and keep consistent discussing the nutrition during each separate section of the event.

In describing the medical conditions of the athlete please elaborate on the fact he had "problems" with weight loss.  What sort of problems?  The authors indicate a "small heat stroke", but by definition heat stroke is a life-threatening condition that typically requires hospitalization and advanced medical care.  Please clarify this diagnosis. The authors continue to describe the "mentally hardest days" but give no indication of how this was measured, please clarify. Additionally after listing injury/illnesses, the authors indicate that no injuries occurred but this should be clarified that no adverse conditions occurred that would preclude participation.

The statistical analysis omits the model that assesses swimming performance as the dependant variable and the factors (weight, ambient conditions, previous days performance, sleep, etc.) that influenced it.  Please reanalyze and include this data.

Section 4.2 describes the aspect of experience which the authors conclude is a determining factor in athlete success.  It would be improved to discuss all of the time of this athlete's personal records and include a statement on his personal best single Ironman-distance race time for comparison. The last two sentence (lines 287-289) are redundant when discussing age.

Section 4.4 discusses HR during the bike and run, but not the swim.  Was HR not taken in the swim?  If it was please include.  Additionally this leads to a question in section 4.4 related to mean EE.  How was this calculated?  If there was no HR taken during the swim, how was this estimated and how valid of a measure is this then?  Also, is the ~4kg weight loss each day or total?  That is unclear in line 315.  What was the average daily loss and recovery of weight.  It would be helpful to discuss post-event weight loss and then weight regain the following morning.  This should also be considered in the analysis. The author's hypothesis in line 340 indicate that intensity was lower than a typical Ironman-distance event, but this athlete's daily finishing times are in line with the average finish times at an Ironman event indicating that there is not a lower intensity during the repeated efforts.  Please eliminate this or rephrase.

A large segment is used to discuss body composition.  This is however inappropriate as the current study has no way to infer the effects on composition as mass was the only variable assessed. Several assumptions are made (lines 346 and 349) and should be eliminated without appropriate data to back it up.

The conclusions of this study fail to indicate the directions of the predictions found. The study also concludes that experience and training volume are indicative of performance however these variable are not part of this study and this statement is out of context and anecdotal here. (lines 371-2)  The paper also needs to evaluate the limitations of the current study, including but not exclusive to missing measures of body composition, the effects of the outlier data (last two days of performance), availability of HR during the swimming event and others if my points in previous sections above are unable to be addressed adequately (e.g., humidity, wind direction, nutritional details, etc.).

Finally, of great concern is the heavy use of self-references for this work.  35 out of the 48 references (73%) are other works from the current authors.  The current study needs to refer to work outside of the same authors for validity and generalizability of the study.  There is significant work done on the physiology and impact of ultra-endurance events that should be included in the introduction and discussion. I do not feel a 73% self-reference rate is appropriate.

Author Response

Reviewer 2

The authors investigate an interesting and novel case-study of a repetitive ultra-endurance performance.  The case presented includes new findings related to pacing, recovery, and impact of physiology and environmental conditions on repeat performance of an Ironman-distance triathlon. We applaud the authors on their ability to compile data from this 40-day effort.

Answer: We thank the expert reviewer for all his comments.

There are several serious concerns that preclude me from recommending this study for publication in its current state.  There are omissions in the methods, discussion and conclusion and a lack of identification of any study limitations. Additionally, significant revision for English language and style is required.  The manuscript as currently written is difficult to read and uses generic adjectives instead of specific language when referring to important segments of the study.

Answer: We expanded the Introduction, we added new statistical analyses, adapted the discussion and conclusion, and inserted the limitations. Where possible we changed the text, but some parts of the manuscript are descriptive in nature with the comments of the athlete. We think this is no fault because we can only rely on the data and comments of the athlete.

The term Ironman triathlon is a corporate brand.  Therefore, as this event was not sanctioned or organized by the WTC Ironman corporation, all instances should be referred to as "Ironman-distance" events throughout the manuscript.

Answer: We agree with the expert reviewer and changed throughout in the manuscript.

Ethically, in section 2.1, the study states that all data was publicly available.  If so, please include the direct link.  I am concerned that data related to heart rate and body weight was however not publicly available and therefore this does not meet the standard for a waiver.

Answer: Section 2.1 was changed to ‘This study was approved by the Institutional Review Board of Kanton St. Gallen, Switzerland, with a waiver of the requirement for informed consent of the participant as the study involved the analysis of publicly available data. Heart rates [https://raitratasepp.ee/blog/40x-ultra-triathlon-statistics-by-disciplines-and-days] and body weight [https://raitratasepp.ee/blog/40-kordse-ultratriatloni-statistika-alade-ja-paevade-loikes] are free available from the blog of the athlete. The study was conducted in accordance with recognized ethical standards according to the Declaration of Helsinki adopted in 1964 and revised in 2013’.

In describing the athlete in section 2.2 a statement is made that all training was "short and fast."  Please clarify this generic description.  Without qualifiers it is difficult to know what short and fast mean.  Please define. Include an average training week for this athlete.

Answer: We expanded that section to ‘All this training units were rather short and fast. In other terms, his training units follow one after the other without a break. With this principle, he saves time and trains the transition from one discipline to the other. By changing disciplines, he also teaches his body to recovery during movements. As an example, he starts his training day with running 11 km, changes to swimming 4.4 km, changes then to running 13 km, followed by cycling for 1.5 hours. He likes to split the distance. In order to run 84.4 km (two marathons) in a Double Iron ultra-triathlon in 6 hours, he needs to run long distances, but by splitting. As an example, when running 64.1 km, he starts with running 16.4 km, after a break of 1 hour, he runs 15.4 km, after a small break he runs 20.3 km, and after a further short break 12 km. With this principle, he can run the same distance a few days later in three (21.7 km, 25.9 km, and 16.3 km) compared to four runs.’ Below is an example of a training week (in Estonian).

In section 2.3 there are conflicting statements regarding whether HR was available or not.  Line 99 indicates the athlete did not check HR but line 102 indicates that HR and EE were monitored.  Please clarify.

Answer: We changed to ‘The energy expenditure was measured using the heart rate method with Garmin Forerunner 935 (www.garmin.com)’.

Add an indication of elevation changes for the run course.

Answer: We add in that section ‘He started from the highest point of the lap and had to complete then 12 full laps. The last half lap consisted in the descent to sea level. He measured the elevation by running two full laps and divided the whole elevation (54 m) by two’ and provide below the changes in elevation of 2 running laps.

A better definition of ambient environmental conditions is needed.  "very windy, mostly sunny and basically without rain" is an obtuse categorization of the weather.  What were the means and ranges for these conditions?  What was the direction of the wind relative to the bike and run course? Consider adding a table.

Answer: By the start of the event it was not the idea to perform a scientific study. The first idea of the athlete was to finish 40 days. And, in case of a further attempt for a longer event or improvement of time, record data to analyze in detail. All work to collect data was done by the athlete and his crew and thanks to the athlete we are now able to analyze his incredible effort! From a practical point of view: the athlete performed laps, a lap means he constantly changes direction, so wind direction changes with each change during the lap. We added in that section ‘The average high temperature in October in Fuerteventura is ~26 °C, and no more than 2 days with little rain can be expected (www.guidetocanaryislands.com/fuerteventura-weather-october). We also added the limitations with ‘Although this case study provides a lot of physiological and technical details, some limitations must be acknowledged. The daily measurement of body composition using a bioelectrical impedance analysis or radiological procedures like DEXA [Med Sci Sports Exerc. 2019 Mar;51(3):556-567. doi: 10.1249/MSS.0000000000001803] or MRI [BMC Med. 2013 May 8;11:122. doi: 10.1186/1741-7015-11-122] would help monitor changes in body composition. The inclusion of the last two days with very slow overall times (outlier) might have had an influence on the analysis of the data. Missing data for humidity [Int J Sports Physiol Perform. 2019 Oct 7:1-7. doi: 10.1123/ijspp.2018-0689], wind direction [Med Sci Sports Exerc. 2006 Apr;38(4):726-34.] and nutrition [Res Sports Med. 2019 Apr-Jun;27(2):166-181] also might have had an influence on the outcome of the analysis.

Additionally, it would be useful to consider relative humidity in the context of its effects on performance.  The location of the event precludes generalization and so specific details of the conditions must be better described.

Answer: We added the limitations with ‘Although this case study provides a lot of physiological and technical details, some limitations must be acknowledged. The daily measurement of body composition using a bioelectrical impedance analysis or radiological procedures like DEXA [Med Sci Sports Exerc. 2019 Mar;51(3):556-567. doi: 10.1249/MSS.0000000000001803] or MRI [BMC Med. 2013 May 8;11:122. doi: 10.1186/1741-7015-11-122] would help monitor changes in body composition. The inclusion of the last two days with very slow overall times (outlier) might have had an influence on the analysis of the data. Missing data for humidity [Int J Sports Physiol Perform. 2019 Oct 7:1-7. doi: 10.1123/ijspp.2018-0689], wind direction [Med Sci Sports Exerc. 2006 Apr;38(4):726-34.] and nutrition [Res Sports Med. 2019 Apr-Jun;27(2):166-181] also might have had an influence on the outcome of the analysis.

While the nutrition section gives an idea of the caloric and macro-nutrient intake of the athlete, term such as "big bowl", "lots of gels", "Kellogg's cereal" are too generic for a single study case study.  What were the quantities involved and the timing of the intake? Line 144 indicates a numeric value but there are no units indicated.  Please correct.  The paragraph on running nutrition also includes a statement on salt intake during the bike (line 145) and a statement of morning intake (line 153).  Please remove from this section and keep consistent discussing the nutrition during each separate section of the event.

Answer: By the start of the event it was not the idea to perform a scientific study. The first idea of the athlete was to finish 40 days. And, in case of a further attempt for a longer event or improvement of time, record data to analyze in detail. All work to collect data was done by the athlete and his crew and thanks to the athlete we are now able to analyze his incredible effort! The athlete and his support crew had not the time to take measurement of food intake. In line 144 we give the unit l (liters) for fluid intake. We try to make sections but follow the daily routine of food intake. In the limitations, we add ‘Missing data for humidity [Int J Sports Physiol Perform. 2019 Oct 7:1-7. doi: 10.1123/ijspp.2018-0689], wind direction [Med Sci Sports Exerc. 2006 Apr;38(4):726-34.] and nutrition [Res Sports Med. 2019 Apr-Jun;27(2):166-181] also might have had an influence on the outcome of the analysis’.

In describing the medical conditions of the athlete please elaborate on the fact he had "problems" with weight loss.  What sort of problems? 

Answer: We added ‘Body mass loss was too fast in the first days and the loss continued without stopping toward the half of the event (see Figure 3C)’.

The authors indicate a "small heat stroke", but by definition heat stroke is a life-threatening condition that typically requires hospitalization and advanced medical care.  Please clarify this diagnosis.

Answer: We changed to ‘At the beginning he had problems with the hot weather (temperature was over 30 °C) and on the 6th day he felt overheated at the end of the bike. He took 25 min in T2 zone to recover using ice and consuming cold drinks.’

The authors continue to describe the "mentally hardest days" but give no indication of how this was measured, please clarify.

Answer: We added ‘He was physically in good conditions and had no muscular problems, However, these days, it took him a lot of willpower to cope with the strong winds while cycling. By the end of these days, he felt himself mentally tired’.

Additionally, after listing injury/illnesses, the authors indicate that no injuries occurred but this should be clarified that no adverse conditions occurred that would preclude participation.

Answer: We shortened that section to ‘The second half of the challenge he had problems with muscles that caused some discomfort during the run. During the 40 days, he suffered from different muscular problems, but no overuse injuries.’

The statistical analysis omits the model that assesses swimming performance as the dependent variable and the factors (weight, ambient conditions, previous days performance, sleep, etc.) that influenced it.  Please reanalyze and include this data.

Answer: The model with as DV was not omitted, we did not include initially because intrinsic data from this split is not available (such as HR). Anyhow, in light of this comment, we agree with the expert reviewer and included this as Model 1. Please see Results and Table 3 for the changes.

Section 4.2 describes the aspect of experience which the authors conclude is a determining factor in athlete success.  It would be improved to discuss all of the time of this athlete's personal records and include a statement on his personal best single Ironman-distance race time for comparison. The last two sentence (lines 287-289) are redundant when discussing age.

Answer: We added ‘His personal best time in a single Ironman is 9:50:30 h:min:s in 2013. Since then, he never competed again in a single Ironman triathlon. We changed that sentence to ‘Furthermore, the athlete is at the best age (36 years) for a fast Ironman triathlon performance since the best age for male athletes is ~35 years for a top performance in an Ironman triathlon [20,21]

Section 4.4 discusses HR during the bike and run, but not the swim.  Was HR not taken in the swim?  If it was please include. 

Answer: During the swim he did not record heart rate for his blog. The watch continuously records heart rate to estimate energy expenditure.

Additionally, this leads to a question in section 4.4 related to mean EE.  How was this calculated?  If there was no HR taken during the swim, how was this estimated and how valid of a measure is this then? 

Answer: During the swim he did not record heart rate for his blog. The watch continuously records heart rate to estimate energy expenditure.

Also, is the ~4kg weight loss each day or total?  That is unclear in line 315.  What was the average daily loss and recovery of weight?  It would be helpful to discuss post-event weight loss and then weight regain the following morning.  This should also be considered in the analysis.

Answer: Body weight measures were taken daily, every morning. Body weight after race day is not available. Thus, weight regain is impossible to be calculated, unfortunately. Information were added in methods and discussion for clarity. Thank you for the comment.

The author's hypothesis in line 340 indicate that intensity was lower than a typical Ironman-distance event, but this athlete's daily finishing times are in line with the average finish times at an Ironman event indicating that there is not a lower intensity during the repeated efforts.  Please eliminate this or rephrase.

Answer: We added ‘This is backed up with the personal best time of 9:50:30 h:min:s and the ~11 hours per day for each Ironman-distance triathlon’.

A large segment is used to discuss body composition.  This is however inappropriate as the current study has no way to infer the effects on composition as mass was the only variable assessed. Several assumptions are made (lines 346 and 349) and should be eliminated without appropriate data to back it up.

Answer: We deleted ‘We may therefore assume that the athlete lost during these 40 days primarily body fat and this loss in body fat was ‘ergogenic’ for both running and overall time. The athlete himself meant that faster running times were partly explained by the fact that he lost body weight and got lighter. Three kg less body mass is a great thing in terms of running.’

The conclusions of this study fail to indicate the directions of the predictions found. The study also concludes that experience and training volume are indicative of performance however these variables are not part of this study and this statement is out of context and anecdotal here. (lines 371-2) 

Answer: We deleted ‘Any athlete intending to achieve a similar performance must be aware that an extensive previous experience and a high training volume are needed’ in the conclusions. The conclusions read now as follows: In summary, a fast running performance is significantly influenced by a low body weight, a fast cycling performance in significantly influenced by low wind speed and a fast overall performance is significantly influenced by both fast running and fast cycling performances. This study could be of assistance for coaches and athletes preparing for similar challenges. More in-depth physiological responses (i.e. sleep quality; inflammation; oxidative stress) to such a challenge with or without a follow-up could be the next step. A highly-trained professional triathlete with extensive previous experience is able to finish daily an Ironman-distance triathlon during 40 days where cycling and running performance are highly predictive for overall time.

The paper also needs to evaluate the limitations of the current study, including but not exclusive to missing measures of body composition, the effects of the outlier data (last two days of performance), availability of HR during the swimming event and others if my points in previous sections above are unable to be addressed adequately (e.g., humidity, wind direction, nutritional details, etc.).

Answer: We agree with the expert reviewer and added the limitations with ‘Although this case study provides a lot of physiological and technical details, some limitations must be acknowledged. The daily measurement of body composition using a bioelectrical impedance analysis or radiological procedures like DEXA [Med Sci Sports Exerc. 2019 Mar;51(3):556-567. doi: 10.1249/MSS.0000000000001803] or MRI [BMC Med. 2013 May 8;11:122. doi: 10.1186/1741-7015-11-122] would help monitor changes in body composition. The inclusion of the last two days with very slow overall times (outlier) might have had an influence on the analysis of the data. Missing data for humidity [Int J Sports Physiol Perform. 2019 Oct 7:1-7. doi: 10.1123/ijspp.2018-0689], wind direction [Med Sci Sports Exerc. 2006 Apr;38(4):726-34.] and nutrition [Res Sports Med. 2019 Apr-Jun;27(2):166-181] also might have had an influence on the outcome of the analysis.

Finally, of great concern is the heavy use of self-references for this work.  35 out of the 48 references (73%) are other works from the current authors.  The current study needs to refer to work outside of the same authors for validity and generalizability of the study.  There is significant work done on the physiology and impact of ultra-endurance events that should be included in the introduction and discussion. I do not feel a 73% self-reference rate is appropriate

Answer: We added new references in Introduction and Discussion.

Reviewer 3 Report

Comments to the Authors

GENERAL COMMENTS REGARDING PAPER

This study has the well done theoretical and practical approaches and it can be instructional for teachers and researchers. However, there are some issues that have not been addressed in the manuscript. Despite such issues if these are addressed adequately it may be acceptable for publication following a second review.

MAJOR COMPULSORY REVISIONS METHODS

INTRODUCTION

MAJOR 01. The gap should be improved. In the first paragraph you talk about it, however it is so important and should be presented clearer for the reader. For example: what this manuscript will contribute to the advance of state of art of the area.

1) Contextualization: you present your research field and highlight the importance of the main area. In short, SHOW THE IMPORTANCE.

2) State of the Art: evidence and recent findings in this area.

In short, SHOW HOW THIS CURRENT KNOWLEDGE IS IN THIS AREA.

3) GAP status: you look for open questions, restrictions or limitations of the area you want to investigate. In short, HIGHLIGHTING THE GAP OF KNOWLEDGE YOU WANT TO COMPLETE.

4) Show the importance of your study: show the evidence and/or practical applications of your study. In short, SHOW WHERE YOUR WORK IS GOING TO GO IN THE LARGE AREA BY FILLING IN THE GAP OF KNOWLEDGE THAT YOU SHOWED AT THE GAP.

5) Purpose: to present your objective. In short, WHAT YOU WILL DO.

MATERIALS AND METHODS

MAJOR 02: pg. 3. “2.3. Event, equipment, support and measurements.” How were performed the measurements? ("overall times, body weight, daily highest temperature, wind speed, energy expenditure, mean heart rate, and sleeping time were recorded"). Provide details and protocols. Insert a figure with all procedures and moments to be clearer for the reader.

DISCUSSION

MAJOR 03: “Table 3". RESULTS

CONCLUSIONS

MAJOR 04: What are the main findings? Practical Applications? Make it clear to the reader in the DISCUSSION and improve your CONCLUSION.

MINOR ESSENTIAL REVISIONS

MINOR 1: pg. 1 – line 43. Typo: “fora” to “for a”.

Author Response

Reviewer 3

Comments to the Authors

GENERAL COMMENTS REGARDING PAPER

This study has the well done theoretical and practical approaches and it can be instructional for teachers and researchers. However, there are some issues that have not been addressed in the manuscript. Despite such issues if these are addressed adequately it may be acceptable for publication following a second review.

Answer: We agree with the expert reviewer and worked on all points.

MAJOR COMPULSORY REVISIONS METHODS

INTRODUCTION

MAJOR 01. The gap should be improved. In the first paragraph you talk about it, however it is so important and should be presented clearer for the reader. For example: what this manuscript will contribute to the advance of state of art of the area.

Answer: We agree with the expert reviewer and expanded the Introduction by adding and/or changing the text with: Ultra-endurance athletes are pushing their limits to go for the ultimate limit in endurance performance [Adv Exp Med Biol. 2007;618:255-76]. In ultra-running, the limit has most probably achieved with crossing a continent [BMC Med. 2012 Jul 19;10:76]. In the ‘Trans Europe Foot Race 2009’, a small sample of a few dozens of ultra-marathoners covered the distance of 4,487 km from South Italy to North Cape [BMC Med. 2012 Jul 19;10:77]. However, the total distance of 4,487 km within 64 days [J Sports Med Phys Fitness. 2019 Oct;59(10):1608-1621] was outperformed by the French ultra-marathoner Patrick Maladin with 10,000 km in 100 days (99 days 4 hours 12 minutes) in 2016 with daily a 100 km ultra-marathon [http://patrickmalandain-ultrarun.com/parcours-10000km]. However, in 2019, he broke his own record while he crossed the United States of America from New York to Los Angeles (4,801 km) within 46 days and Canada from Vancouver to Halifax (5,931 km) within 56 days to complete the overall distance of 10,732 km within 102 days 18 hours and 48 minutes [http://patrickmalandain-ultrarun.com/#us]. Similarly, triathletes seem to push also their limits with completing daily an Ironman-distance triathlon for as many days as possible. While the longest scientifically verified self-paced event was 33 Ironman-distance triathlons in 33 days in summer 2013 [7], we present here the pacing in the first and only athlete to complete a scientifically verified self-paced world record attempt to finish 40 Ironman-distance triathlons within 40 days held in autumn 2019. We also investigated potential predictor variables for split disciplines (i.e. cycling and running) and overall performance.

1) Contextualization: you present your research field and highlight the importance of the main area. In short, SHOW THE IMPORTANCE.

Answer: We agree with the expert reviewer and expanded the Introduction by adding and/or changing the text with: Ultra-endurance athletes are pushing their limits to go for the ultimate limit in endurance performance [Adv Exp Med Biol. 2007;618:255-76]. In ultra-running, the limit has most probably achieved with crossing a continent [BMC Med. 2012 Jul 19;10:76]. In the ‘Trans Europe Foot Race 2009’, a small sample of a few dozens of ultra-marathoners covered the distance of 4,487 km from South Italy to North Cape [BMC Med. 2012 Jul 19;10:77]. However, the total distance of 4,487 km within 64 days [J Sports Med Phys Fitness. 2019 Oct;59(10):1608-1621] was outperformed by the French ultra-marathoner Patrick Maladin with 10,000 km in 100 days (99 days 4 hours 12 minutes) in 2016 with daily a 100 km ultra-marathon [http://patrickmalandain-ultrarun.com/parcours-10000km]. However, in 2019, he broke his own record while he crossed the United States of America from New York to Los Angeles (4,801 km) within 46 days and Canada from Vancouver to Halifax (5,931 km) within 56 days to complete the overall distance of 10,732 km within 102 days 18 hours and 48 minutes [http://patrickmalandain-ultrarun.com/#us]. Similarly, triathletes seem to push also their limits with completing daily an Ironman-distance triathlon for as many days as possible. While the longest scientifically verified self-paced event was 33 Ironman-distance triathlons in 33 days in summer 2013 [7], we present here the pacing in the first and only athlete to complete a scientifically verified self-paced world record attempt to finish 40 Ironman-distance triathlons within 40 days held in autumn 2019. We also investigated potential predictor variables for split disciplines (i.e. cycling and running) and overall performance.

2) State of the Art: evidence and recent findings in this area.

In short, SHOW HOW THIS CURRENT KNOWLEDGE IS IN THIS AREA.

Answer: We agree with the expert reviewer and expanded the Introduction by adding and/or changing the text with: Ultra-endurance athletes are pushing their limits to go for the ultimate limit in endurance performance [Adv Exp Med Biol. 2007;618:255-76]. In ultra-running, the limit has most probably achieved with crossing a continent [BMC Med. 2012 Jul 19;10:76]. In the ‘Trans Europe Foot Race 2009’, a small sample of a few dozens of ultra-marathoners covered the distance of 4,487 km from South Italy to North Cape [BMC Med. 2012 Jul 19;10:77]. However, the total distance of 4,487 km within 64 days [J Sports Med Phys Fitness. 2019 Oct;59(10):1608-1621] was outperformed by the French ultra-marathoner Patrick Maladin with 10,000 km in 100 days (99 days 4 hours 12 minutes) in 2016 with daily a 100 km ultra-marathon [http://patrickmalandain-ultrarun.com/parcours-10000km]. However, in 2019, he broke his own record while he crossed the United States of America from New York to Los Angeles (4,801 km) within 46 days and Canada from Vancouver to Halifax (5,931 km) within 56 days to complete the overall distance of 10,732 km within 102 days 18 hours and 48 minutes [http://patrickmalandain-ultrarun.com/#us]. Similarly, triathletes seem to push also their limits with completing daily an Ironman-distance triathlon for as many days as possible. While the longest scientifically verified self-paced event was 33 Ironman-distance triathlons in 33 days in summer 2013 [7], we present here the pacing in the first and only athlete to complete a scientifically verified self-paced world record attempt to finish 40 Ironman-distance triathlons within 40 days held in autumn 2019. We also investigated potential predictor variables for split disciplines (i.e. cycling and running) and overall performance.

3) GAP status: you look for open questions, restrictions or limitations of the area you want to investigate. In short, HIGHLIGHTING THE GAP OF KNOWLEDGE YOU WANT TO COMPLETE.

Answer: We agree with the expert reviewer and expanded the Introduction by adding and/or changing the text with: Ultra-endurance athletes are pushing their limits to go for the ultimate limit in endurance performance [Adv Exp Med Biol. 2007;618:255-76]. In ultra-running, the limit has most probably achieved with crossing a continent [BMC Med. 2012 Jul 19;10:76]. In the ‘Trans Europe Foot Race 2009’, a small sample of a few dozens of ultra-marathoners covered the distance of 4,487 km from South Italy to North Cape [BMC Med. 2012 Jul 19;10:77]. However, the total distance of 4,487 km within 64 days [J Sports Med Phys Fitness. 2019 Oct;59(10):1608-1621] was outperformed by the French ultra-marathoner Patrick Maladin with 10,000 km in 100 days (99 days 4 hours 12 minutes) in 2016 with daily a 100 km ultra-marathon [http://patrickmalandain-ultrarun.com/parcours-10000km]. However, in 2019, he broke his own record while he crossed the United States of America from New York to Los Angeles (4,801 km) within 46 days and Canada from Vancouver to Halifax (5,931 km) within 56 days to complete the overall distance of 10,732 km within 102 days 18 hours and 48 minutes [http://patrickmalandain-ultrarun.com/#us]. Similarly, triathletes seem to push also their limits with completing daily an Ironman-distance triathlon for as many days as possible. While the longest scientifically verified self-paced event was 33 Ironman-distance triathlons in 33 days in summer 2013 [7], we present here the pacing in the first and only athlete to complete a scientifically verified self-paced world record attempt to finish 40 Ironman-distance triathlons within 40 days held in autumn 2019. We also investigated potential predictor variables for split disciplines (i.e. cycling and running) and overall performance.

 4) Show the importance of your study: show the evidence and/or practical applications of your study. In short, SHOW WHERE YOUR WORK IS GOING TO GO IN THE LARGE AREA BY FILLING IN THE GAP OF KNOWLEDGE THAT YOU SHOWED AT THE GAP.

Answer: We agree with the expert reviewer and expanded the Introduction by adding and/or changing the text with: Ultra-endurance athletes are pushing their limits to go for the ultimate limit in endurance performance [Adv Exp Med Biol. 2007;618:255-76]. In ultra-running, the limit has most probably achieved with crossing a continent [BMC Med. 2012 Jul 19;10:76]. In the ‘Trans Europe Foot Race 2009’, a small sample of a few dozens of ultra-marathoners covered the distance of 4,487 km from South Italy to North Cape [BMC Med. 2012 Jul 19;10:77]. However, the total distance of 4,487 km within 64 days [J Sports Med Phys Fitness. 2019 Oct;59(10):1608-1621] was outperformed by the French ultra-marathoner Patrick Maladin with 10,000 km in 100 days (99 days 4 hours 12 minutes) in 2016 with daily a 100 km ultra-marathon [http://patrickmalandain-ultrarun.com/parcours-10000km]. However, in 2019, he broke his own record while he crossed the United States of America from New York to Los Angeles (4,801 km) within 46 days and Canada from Vancouver to Halifax (5,931 km) within 56 days to complete the overall distance of 10,732 km within 102 days 18 hours and 48 minutes [http://patrickmalandain-ultrarun.com/#us]. Similarly, triathletes seem to push also their limits with completing daily an Ironman-distance triathlon for as many days as possible. While the longest scientifically verified self-paced event was 33 Ironman-distance triathlons in 33 days in summer 2013 [7], we present here the pacing in the first and only athlete to complete a scientifically verified self-paced world record attempt to finish 40 Ironman-distance triathlons within 40 days held in autumn 2019. We also investigated potential predictor variables for split disciplines (i.e. cycling and running) and overall performance.

5) Purpose: to present your objective. In short, WHAT YOU WILL DO.

Answer: We agree with the expert reviewer and expanded the Introduction by adding and/or changing the text with: Ultra-endurance athletes are pushing their limits to go for the ultimate limit in endurance performance [Adv Exp Med Biol. 2007;618:255-76]. In ultra-running, the limit has most probably achieved with crossing a continent [BMC Med. 2012 Jul 19;10:76]. In the ‘Trans Europe Foot Race 2009’, a small sample of a few dozens of ultra-marathoners covered the distance of 4,487 km from South Italy to North Cape [BMC Med. 2012 Jul 19;10:77]. However, the total distance of 4,487 km within 64 days [J Sports Med Phys Fitness. 2019 Oct;59(10):1608-1621] was outperformed by the French ultra-marathoner Patrick Maladin with 10,000 km in 100 days (99 days 4 hours 12 minutes) in 2016 with daily a 100 km ultra-marathon [http://patrickmalandain-ultrarun.com/parcours-10000km]. However, in 2019, he broke his own record while he crossed the United States of America from New York to Los Angeles (4,801 km) within 46 days and Canada from Vancouver to Halifax (5,931 km) within 56 days to complete the overall distance of 10,732 km within 102 days 18 hours and 48 minutes [http://patrickmalandain-ultrarun.com/#us]. Similarly, triathletes seem to push also their limits with completing daily an Ironman-distance triathlon for as many days as possible. While the longest scientifically verified self-paced event was 33 Ironman-distance triathlons in 33 days in summer 2013 [7], we present here the pacing in the first and only athlete to complete a scientifically verified self-paced world record attempt to finish 40 Ironman-distance triathlons within 40 days held in autumn 2019. We also investigated potential predictor variables for split disciplines (i.e. cycling and running) and overall performance.

MATERIALS AND METHODS

MAJOR 02: pg. 3. “2.3. Event, equipment, support and measurements.” How were performed the measurements? ("overall times, body weight, daily highest temperature, wind speed, energy expenditure, mean heart rate, and sleeping time were recorded"). Provide details and protocols. Insert a figure with all procedures and moments to be clearer for the reader.

Answer: By the start of the event it was not the idea to perform a scientific study. The first idea of the athlete was to finish 40 days. And, in case of a further attempt for a longer event or improvement of time, record data to analyze in detail. All work to collect data was done by the athlete and his crew and thanks to the athlete we are now able to analyze his incredible effort! All details are presented in the section 2.3. The athlete has no further details to provide and followed no specific protocol as it was not intended to become a scientific study.

DISCUSSION

MAJOR 03: “Table 3". RESULTS

Answer: We changed to ‘Table 3 presents the results of the four multivariate regression models. The first multivariate regression model used swimming performance as the dependent variable and the following variables as the independent variables: sleep duration; body weight; wind speed; daily highest temperature; and overall time in the previous day. The model showed statistical significance (p=0.010) with an adjusted R2 of 0.28. the only significant independent variables to influence cycling performance was wind speed (p=0.005), and a trend (p=0.062) for body weight. The second multivariate regression model used cycling performance as the dependent variable and the following variables as the independent variables: swimming time; sleep duration; body weight; wind speed; daily highest temperature; average cycling heart rate, and overall time in the previous day. The model showed statistical significance (p<0.001) with an adjusted R2 of 0.78. Wind speed, overall time in the previous day and average heart race during cycling were the significant independent variables (p<0.05) to influence cycling performance.

 CONCLUSIONS

MAJOR 04: What are the main findings? Practical Applications? Make it clear to the reader in the DISCUSSION and improve your CONCLUSION.

Answer: We inserted a section with practical application ‘This case study shows that it is possible to perform an Ironman triathlon daily for 40 days. The findings in this study may help any athlete intending to outperform the performance with a faster time per Ironman or to finish more Ironman triathlons’ and changed the conclusions to ‘In summary, a fast running performance is significantly influenced by a low body weight, a fast cycling performance in significantly influenced by low wind speed and a fast overall performance is significantly influenced by both fast running and fast cycling performances. This study could be of assistance for coaches and athletes preparing for similar challenges. More in-depth physiological responses (i.e. sleep quality; inflammation; oxidative stress) to such a challenge with or without a follow-up could be the next step. A highly-trained professional triathlete with extensive previous experience is able to finish daily an Ironman-distance triathlon during 40 days where cycling and running performance are highly predictive for overall time.

MINOR ESSENTIAL REVISIONS

MINOR 1: pg. 1 – line 43. Typo: “fora” to “for a”.

Answer: We thank the expert reviewer for finding this error and we changed as suggested.

Round 2

Reviewer 2 Report

I thank the authors for their revisions to date.  The manuscript is much improved. The additional data, analysis and clarifications provide much greater coherency to the description of the study. There are some minor and specific recommendations and clarification questions that follow that will improve the manuscript further.

There are still several instances of referring to these events as Ironman rather than Ironman-distance (e.g. discussion section 4.1, 4.9.  Please correct.

Line 66: grammatical editing needed.

Line 69: define what is meant by scientifically verified.  This manuscript appears to utilize self-reported data rather than scientifically verified.

Section 2.2: It would be useful to include the Ironman-distance personal best in this section rather than later.

Line 104: The sentence referring to "short and fast" training remains too subjective.  What may be short to an ultra-endurance athlete, certainly is not short in the typical context of exercise training.  The term "fast" is likewise inappropriate as no context of exercise intensity is present.

Line 105:  Would transition be a better term than "follow one after the other with no break"?

Line 107: Grammatical editing needed.

Line 118: Does specific strategy refer to pacing?  This is unlcear.

Line 118-120: This sentence is ambiguous and requires clarification. What does chasing the fastest times indicate?  Was he following a pacer during the event? what does taking a maximum refer to?

Line 125-6: The sentence "Over the years..." is subjective and should be removed.

Line 136-9:  The addition of elevation change for the run course is appreciated.  However, as currently written is is difficult to decipher the true elevation changes.  Is 54m the total elevation change per lap? Please revise.

Line 143: Please indicate the time change is related to daylight savings time as there are international readers that may be unfamiliar with the reason for changing the clock an hour.

Lines 144-8:  It is unclear if the daily high temperatures and wind speeds were utilized for environmental analysis.  Was it the daily high or the high during each segment of the event? (e.g., high winds during the swim might change drastically by the time the cycle event starts and would therefore have little to no impact).  Please clarify and if daily high temperatures and wind speeds were used, this should be noted in the limitations.

Line 154: Do the authors mean laps?

Line 160 & 181: drank vs. drunk

Line 170-1: Grammatical editing needed.

Line 192: Please indicate that the mentally "hardest" days are self-report from the athlete and not utilizing a standard measurement.  Additionally, line 195 should indicate that he self-reported as being mentally tired.

Line 202-3: This sentence is ambiguous and seems unnecessary.

Line 220: No need for the 5% value.  Simple state that significance was set at p<0.05

Figure 1 & 2.  The captions may be reversed for these figures as 1 seems to indicate 10-day averages and 2 seems to indicate individual points for each event. For figure one, the caption should also indicate what the (*) means.

Figure 3B: The points are difficult to differentiate in this figure.  Because they are intertwined it is difficult to read and I suggest using different shading in addition to different shapes.

Line 249:  Should this read that the times of the event increased (got slower) until day 15, improved to day 31 and then again increased until the end indicating a sinusoidal pattern?

Line 266:  Should this refer to the swimming model rather than the cycling model?

Lines 262-85:  The text is redundant with the table.  It would be helpful to just refer to the table and remove listing of each variable in each model. Ideally only point out the items of significance.

Lines 287-92: There should be a mention that wind speed  also was predictive of the swimming segment. It is missing in this summative paragraph.

Table 3: The caption fails to mention the swimming model included in the data.

Line 330-1: The indication of the "best age" is slightly misleading.  Do the authors mean "most competitive"?

Line 339: delete "only"

Line 365: This statement would be better stated that it is evidenced due to this specific athlete's daily performance as compared to their personal best in a single Ironman-distance event.

Line 393: Recognize that wind-speed was also predictive for the swim performance.

Line 397: The idea that "most" triathletes can cope with the heat isn't supported, please either provide an appropriate reference for this statement or remove.

Section 4.7: This section doesn't add any additional information to the manuscript.  The term "best way" is subjective and ambiguous. Please consider removing the section as a whole.

Section 4.8: It would be helpful to identify the limitations of when the environmental measures were made.  Were they just daily highs or race-segment specific?

Line 417: Would it be more appropriate to refer here to finish more "consecutive" Ironman distance triathlons?

Line 421: It seems that cycling performance is influenced also by the previous day's effort?

Line 424: I recommend including body composition measures in the list of more in-depth physiologic measures for consistency with the paper.

Line 426: This would be more clear if it indicated the authors were referring to triathlons on consecutive 40 days.

Author Response

I thank the authors for their revisions to date.  The manuscript is much improved. The additional data, analysis and clarifications provide much greater coherency to the description of the study. There are some minor and specific recommendations and clarification questions that follow that will improve the manuscript further.

Answer: We thank the expert reviewer for his/her comments and worked again on the manuscript.

There are still several instances of referring to these events as Ironman rather than Ironman-distance (e.g. discussion section 4.1, 4.9.  Please correct.

Answer: We changed in 4.1 and 4.9. However, in 4.1, the mentioned studies investigated official Ironman races. But no problem for us to chance also here.

Line 66: grammatical editing needed.

Answer: We changed to ‘Triathletes push their limits as much as the ultra-runners with completing daily an Ironman-distance triathlon for as many days as possible’.

Line 69: define what is meant by scientifically verified.  This manuscript appears to utilize self-reported data rather than scientifically verified.

Answer: One of the co-authors is a triathlete competing since more than 25 years in official Ironman races and longer triathlon distances. He knows many ultra-triathletes and ultra-runners pushing their limits to the edge. There are several athletes who announce to have set a new record, see for example http://ultratriathlet.blogspot.com/2019/01/route-66-die-challenge.html or https://www.redbull.com/car-en/the-iron-cowboy-did-50-marathons-in-50-states-in-50-day. However, these athletes have not offered their data and therefore their ‘records’ are difficult to verify. We changed that sentence to ‘While the longest scientifically verified self-paced event was 33 Ironman-distance triathlons in 33 days in summer 2013 [7], we present here the pacing in the first and only athlete to complete a scientifically verified (all data collected and available for detailed analysis) self-paced world record attempt to finish 40 Ironman-distance triathlons within 40 days held in autumn 2019’.

Our group has analyzed also other self-paced world record attempts, see Int J Environ Res Public Health. 2019 Aug 16;16(16). pii: E2943. doi: 10.3390/ijerph16162943, Springerplus. 2014 May 28;3:269. doi: 10.1186/2193-1801-3-269. eCollection 2014 or Springerplus. 2015 Oct 29;4:650. doi: 10.1186/s40064-015-1445-1. eCollection 2015 where the athletes have offered all their collected data and were interested in analysis. So, the other ‘records’ of 50 and 66 Ironman triathlons in 50 and 66 days could also be fake. Because no data were available to check the performance.

Furthermore, one of the co-authors knows the investigated athlete since years personally. The athlete is for sure among the most professional and most dedicated ultra-triathletes in the present times!

Section 2.2: It would be useful to include the Ironman-distance personal best in this section rather than later.

Answer: We added as suggested in section 2.2.

Line 104: The sentence referring to "short and fast" training remains too subjective.  What may be short to an ultra-endurance athlete, certainly is not short in the typical context of exercise training.  The term "fast" is likewise inappropriate as no context of exercise intensity is present.

Answer: We changed to ‘He developed his own training principle where his training units follow one after the other without a break’

Line 105:  Would transition be a better term than "follow one after the other with no break"?

Answer: We changed to ‘He developed his own training principle where his training units have short transitions’.

Line 107: Grammatical editing needed.

Answer: We deleted since it cannot be described in other way.

Line 118: Does specific strategy refer to pacing?  This is unclear.

Answer: We changed to ‘The athlete followed no specific pacing strategy’.

Line 118-120: This sentence is ambiguous and requires clarification. What does chasing the fastest times indicate?  Was he following a pacer during the event? what does taking a maximum refer to?

Answer: We changed to ‘Every day of the event, he tried to swim, cycle and run as fast as possible with the consideration that he must be able to compete the following day in the same manner’.

Line 125-6: The sentence "Over the years..." is subjective and should be removed.

Answer: We changed to ‘Based upon his previous experience he knows himself very well, and most of the time he can notice and understand these signals’.

Line 136-9:  The addition of elevation change for the run course is appreciated.  However, as currently written is difficult to decipher the true elevation changes.  Is 54m the total elevation change per lap? Please revise.

Answer: We changed to ‘He measured the elevation by running two full laps and divided the whole elevation (54 m) by two resulting in 27 m elevation difference per running lap’.

Line 143: Please indicate the time change is related to daylight savings time as there are international readers that may be unfamiliar with the reason for changing the clock an hour.

Answer: We added ‘This procedure was chosen in order to be able to run as long as possible in daylight due to the change from summer to winter time’.

Lines 144-8:  It is unclear if the daily high temperatures and wind speeds were utilized for environmental analysis.  Was it the daily high or the high during each segment of the event? (e.g., high winds during the swim might change drastically by the time the cycle event starts and would therefore have little to no impact).  Please clarify and if daily high temperatures and wind speeds were used, this should be noted in the limitations.

Answer: We added in the limitations ‘We only used the daily maximum of air temperature and wind speed which might have had an influence on our analysis’.

Line 154: Do the authors mean laps?

Answer: We changed to ‘In swimming, he had always two persons where one was counting the laps.’

Line 160 & 181: drank vs. drunk

Answer: We changed as suggested.

Line 170-1: Grammatical editing needed.

Answer: We changed to ‘On the run course he stopped only for comfort breaks and changes of shirts or running shoes.’

Line 192: Please indicate that the mentally "hardest" days are self-report from the athlete and not utilizing a standard measurement. 

Answer: We changed to ‘The mentally hardest days (self-reported by the athlete and not utilizing a standard measurement) were days 6, 12 and 18 and were mainly caused by very difficult wind conditions (i.e. strength or direction)’.

Additionally, line 195 should indicate that he self-reported as being mentally tired.

Answer: We changed to ‘By the end of these days, he felt himself mentally tired (self-reported by the athlete and not utilizing a standard measurement)’.

Line 202-3: This sentence is ambiguous and seems unnecessary.

Answer: We deleted as suggested

Line 220: No need for the 5% value.  Simple state that significance was set at p<0.05

Answer: We changed to ‘The statistical significance was set at p≤ 0.05’.

Figure 1 & 2.  The captions may be reversed for these figures as 1 seems to indicate 10-day averages and 2 seems to indicate individual points for each event. For figure one, the caption should also indicate what the (*) means.

Answer: We reversed the captions and add the meaning of “*” in Figure 1, as requested.

Figure 3B: The points are difficult to differentiate in this figure.  Because they are intertwined it is difficult to read and I suggest using different shading in addition to different shapes.

Answer: We fixed Figure 3B to make easier to differentiate.

Line 249:  Should this read that the times of the event increased (got slower) until day 15, improved to day 31 and then again increased until the end indicating a sinusoidal pattern?

Answer: We changed to ‘Times of the event increased (got slower) until day 15, improved to day 31 and then again increased until the end indicating a sinusoidal pattern’.

Line 266:  Should this refer to the swimming model rather than the cycling model?

Answer: We changed to ‘The first model (swimming model) showed statistical significance (p=0.010) with an adjusted R2 of 0.28’.

Lines 262-85:  The text is redundant with the table.  It would be helpful to just refer to the table and remove listing of each variable in each model. Ideally only point out the items of significance.

Answer: We changed to ‘presents the results of the four multivariate regression models. The first multivariate regression model (swimming model) used swimming performance as the dependent variable and showed statistical significance (p=0.010) with an adjusted R2 of 0.28. The only significant independent variables to influence cycling performance was wind speed (p=0.005), and a trend (p=0.062) for body weight. The second multivariate regression model (cycling model) used cycling performance as the dependent variable and showed statistical significance (p<0.001) with an adjusted R2 of 0.78. Wind speed, overall time in the previous day and average heart race during cycling were the significant independent variables (p<0.05) to influence cycling performance. The third multivariate regression model (running model) used running time as the dependent variable and showed statistical significance (p<0.001) with an adjusted R2 of 0.69. Body weight and average heart rate during running were the significant independent variables (p<0.05) to influence running performance. The fourth multivariate regression model (overall model) used overall time as the dependent variable and showed statistical significance (p<0.001) with an adjusted R2 of 0.98. Cycling performance, running performance and body weight were the significant independent variables (p<0.05) to influence overall time’.

Lines 287-92: There should be a mention that wind speed also was predictive of the swimming segment. It is missing in this summative paragraph.

Answer: We changed to ‘In this self-paced world record attempt, we found that (i) performances in cycling and running were predictive for overall time, (ii) overall time was predictive for cycling performance of the following day, (iii) wind speed was predictive for swimming and cycling time, (iv) body mass was predictive for running and overall time, (v) heart rate was predictive for cycling and running time, (vi) temperature and sleep time were not predictive to performance, and (vii) the highest variation occurred in the marathon split times’.

Table 3: The caption fails to mention the swimming model included in the data.

Answer: We changed to ‘Results of the multivariate regression models to test which variable has the highest influence in swimming, cycling, running and overall time in 40 Ironman-distance triathlons in 40 days.’

Line 330-1: The indication of the "best age" is slightly misleading.  Do the authors mean "most competitive"?

Answer: We changed to ‘Furthermore, the athlete is at the best age (36 years) for a fast Ironman triathlon performance since the most competitive age for male athletes is ~35 years for a top performance in an Ironman triathlon’.

Line 339: delete "only"

Answer: We deleted as suggested.

Line 365: This statement would be better stated that it is evidenced due to this specific athlete's daily performance as compared to their personal best in a single Ironman-distance event.

Answer: We changed to ‘This is evidenced due to this specific athlete's daily performance as compared to their personal best in a single Ironman-distance event’.

Line 393: Recognize that wind-speed was also predictive for the swim performance.

Answer: We changed to ‘It was shown that wind speed was predictive for swimming and cycling, but the daily highest temperature was not related to performance’.

Line 397: The idea that "most" triathletes can cope with the heat isn't supported, please either provide an appropriate reference for this statement or remove.

Answer: We deleted that statement.

Section 4.7: This section doesn't add any additional information to the manuscript.  The term "best way" is subjective and ambiguous. Please consider removing the section as a whole.

Answer: We changed as suggested.

Section 4.8: It would be helpful to identify the limitations of when the environmental measures were made.  Were they just daily highs or race-segment specific?

Answer: We changed to ‘We only used the daily maximum of air temperature and wind speed which might have had an influence on our analysis’.

Line 417: Would it be more appropriate to refer here to finish more "consecutive" Ironman distance triathlons?

Answer: We changed to ‘The findings in this study may help any athlete intending to outperform the performance with a faster time per Ironman or to finish more consecutive Ironman-distance triathlons.’

Line 421: It seems that cycling performance is influenced also by the previous day's effort?

Answer: We changed to ‘In summary, a fast running performance is significantly influenced by a low body weight, a fast cycling performance in significantly influenced by both a low wind speed and by the previous day's effort and a fast overall performance is significantly influenced by both fast running and fast cycling performances’.

Line 424: I recommend including body composition measures in the list of more in-depth physiologic measures for consistency with the paper.

Answer: We changed to ‘More in-depth physiological responses (i.e. body composition measures, sleep quality, inflammation, oxidative stress) to such a challenge with or without a follow-up could be the next step’.

Line 426: This would be clearer if it indicated the authors were referring to triathlons on consecutive 40 days.

Answer: We changed to ‘A highly-trained professional triathlete with extensive previous experience is able to finish daily an Ironman-distance triathlon on consecutive 40 days where cycling and running performance are highly predictive for overall time’.